# Re-examining the effects of drought on intimate-partner violence

**Matthew Cooper**[1,2]*, **Austin Sandler**[1], **Sveva Vitellozzi**[3], **Yeyoung Lee**[4], **Greg Seymour**[5], **Beliyou Haile**[5], **Carlo Azzari**[5]

**1** Department of Geographical Sciences, University of Maryland, College Park, Maryland, United States of America, **2** T.H. Chan School of Public Health, Harvard University, Brookline, Massachusetts, United States of America, **3** Department of Economics, University of Florence, Florence, Italy, **4** Department of Agricultural, Food, and Resource Economics, Michigan State University, East Lansing, Michigan, United States of America, **5** International Food Policy Research Institute, Washington, D.C., United States of America

* mcooper@hsph.harvard.edu

**Data Availability Statement:** The data on intimate partner violence cannot be shared publicly as it contains sensitive and potentially identifying patient information, and access requires the approval of the Demographic and Health Surveys Program

## Abstract

Droughts are associated with several societal ills, especially in developing economies that rely on rainfed agriculture. Recently, researchers have begun to examine the effect of droughts on the risk of Intimate-Partner Violence (IPV), but so far this work has led to inconclusive results. For example, two large recent studies analyzed comparable data from multiple sub-Saharan African countries and drew opposite conclusions. We attempt to resolve this apparent paradox by replicating previous analyses with the largest data set yet assembled to study drought and IPV. Integrating the methods of previous studies and taking particular care to control for spatial autocorrelation, we find little association between drought and most forms of IPV, although we do find evidence of associations between drought and women's partners exhibiting controlling behaviors. Moreover, we do not find significant heterogeneous effects based on wealth, employment, household drinking water sources, or urban-rural locality.

## Introduction

According to the Violence Against Women and Girls Series commissioned by *The Lancet*, "The elimination of violence against women and girls is central to equitable and sustainable social and economic development" [1]. Currently, intimate partner violence, or IPV, is a major source of physical and psychological suffering for women all over the world, affecting an estimated 30% of women globally, with higher rates in developing countries [2]. The Centers for Disease Control and Prevention defines IPV as physical violence, sexual violence, stalking, or psychological harm by a current or former partner or spouse [3].

Some scholars have proposed that drought may increase a woman's probability of experiencing IPV. Climate shocks can have a variety of impacts on human well-being, and many of these impacts more severely effect women. For example, in many contexts women farm different crops with different inputs than men [4–6]. And women's welfare is more affected by weather variability [7, 8] and droughts [9], especially in arid and semi-arid regions

(DHS Program). Researchers can request access to this data from the DHS Program at dhsprogram. org. All other datasets used are freely available and documented in the article. Code to replicate the findings can be found at https://github.com/mcooper/gbv.

**Funding:** MC received grant 2019X080.COO from the International Food Policy Research Institute for work in this paper (ifpri.org). Scientists from IFPRI played a role in the study design, data collection and analysis, decision to publish, and preparation of the manuscript.

**Competing interests:** We have no competing interests to disclose.

[10, 11]. Given this robust body of research on gender specific climate impacts, some suspect a further linkage between women's vulnerability to IPV and climate shocks, specifically drought. This is an especially urgent question given that climate change is expected to increase the frequency and severity of drought in the coming years [12].

Two recent studies using different methodologies have investigated this hypothesis, but disagreed on whether or not there is an observable relationship between drought and IPV [13, 14]. In this study, we synthesize the methodologies presented in the two previous studies and draw on an even larger sample from sub-Saharan Africa (SSA), South and Southeast Asia (Asia), and Latin America and the Caribbean (LAC) to explore linkages between drought and the risk of IPV. We examine the effects of drought on four forms of IPV: controlling behaviors, emotional violence, physical violence, and sexual violence. In our models, we take care to control for spatial autocorrelation, which can introduce measurement errors that result in biased and inconsistent parameter estimates due primarily to non-random exposure to drought [15].

This paper is organized as follows. The rest of this introduction discusses potential pathways by which drought could affect IPV and relevant empirical literature; the "Materials and Methods" describes the data used in the analyses and outlines our identification strategy; the "Results" section presents regression results; and the "Discussion" discusses the implications of our study's findings.

## Pathways from drought to IPV

One pathway by which drought may increase women's vulnerability to IPV is through asymmetrical impacts of drought on women. Drought has major impacts on rainfed agriculture, and in many cases, women farmers have less access to extension services, agricultural inputs, and financial credit than men [16–18] and are thus disproportionately affected by drought. Additionally, many of the adaptation strategies that are available to women require significant time investments, which may be infeasible for women who have household and childcare tasks [19].

Much of the research shows how the impacts of climate shocks disproportionately affect women in social contexts where they are already marginalized and vulnerable. For example, droughts increase the probability of early marriage in Africa, where women have limited power to make decisions about who they marry [20] and in Malawi, temperature shocks more adversely affect household welfare among households where women solely managed the land [21]. In terms of exposure to weather-related events, studies show that women are more vulnerable to weather variability than men because of the gender roles embedded in socio-cultural norms and institutional contexts [21, 22]. For example, a study in India has shown that gender inequalities decrease women's decision-making authority to cope with weather variability [23].

Relevant to the analysis of drought and IPV, a specific body of the literature focuses on the effects of water insecurity on mental health. Studies from parts of Asia [24], Africa [25], and Latin America [26] have shown that water insecurity is associated with worse mental health, and other studies have shown that there a number of pathways through which water insecurity can affect mental health and well-being [27, 28]. While these studies have primarily focused on women's mental health, an analysis from Uganda showed that water insecurity is more strongly associated with depression in men than in women, with implications for IPV [29]. Moreover, food security, which is often induced by drought, has been shown to be associated with the risk of IPV in Kenya and Nepal [30, 31].

A large literature demonstrates that changes in a woman's income, or her control over resources more generally, can affect her risk of experiencing IPV [32–35]. One set of related theories from sociology and psychology link IPV to poverty-related stress and the emotional

well-being of household members, whereby any improvement in household economic situation leads to a reduction in IPV [36–39]. Economists instead tend to focus on relative changes in women's resources using non-cooperative or other variants of collective household bargaining models [40–42]. Within this family of models, the predicted association between women's income and IPV depends on the mode of violence. If men use violence to express frustration, an increase in women's income decreases IPV by improving her threat point and thus bargaining power within the relationship [43]. If men use violence as an instrument to control women's behavior [44, 45] or extract resources from women [46], an increase in women's income may increase IPV. In contrast to instrumental or extractive theories of IPV, recent evidence from several studies of cash transfer programs supports the view that improvements in women's income most often decreases IPV [47–49].

Additionally, the stressors associated with drought may increase rates of violence against women even when women are not disproportionately affected by the economic and agricultural impacts of drought. A large body of research has shown that droughts and other shocks are associated with increased rates of violence generally [50], including civil conflict [51, 52] and crime [53]. Psychology has long observed that a frustrating emotional event increases aggressive and violent behavior in some individuals [54, 55]. This is related to insights around how poverty and undernourishment affect cognitive function and self control [39], which is relevant given that droughts have been shown to affect nutrition [56–58] and income [59–61]. Moreover, the direct stresses associated with drought conditions may increase rates of IPV. For example, heat waves, which commonly co-occur with drought [62], have been associated with IPV in Spain [63].

However, it is also possible that there are no pathways leading from drought to IPV at continental scales. Firstly, the direct impacts of drought on agricultural production and income vary significantly across agro-ecological and socioeconomic contexts [64]. Secondly, the relationship between income shocks and IPV varies across cultures though complex and context-specific pathways [65]. While in most cases a decrease in a woman's relative income is associated with increased IPV, in other cases, a decrease in a male's relative income can also increase rates of IPV, by provoking a crisis of male identity [66]. This heterogeneity in the relationship between women's income and IPV is illustrated by a study from 28 countries that found both positive and negative relationships between a women's asset ownership and her risk of IPV, depending on the country [67]. Thus, both the first-order impacts of drought on agriculture and income as well as the second-order impacts on IPV are highly dependent on local contexts and may not yield consistent patterns when data is analyzed at continental scales.

## Review of current evidence

Prior to 2020, much of the evidence for an association between drought and IPV was in discussion papers and conference proceedings, often using a single cross-sectional survey from one country. In a published extended abstract for the Population Association of America, Sahni and Sihna looked at IPV in India in 2011–2012, and report that households experiencing a wetter-than-average year were less likely to report IPV being common in their communities [68]. Similarly, in a discussion paper from the Institute of Labor Economics (IZA), Abiona and Koppensteiner use data from the 2008–2009 Tanzania LSMS-ISA (Living Standards Measurement Survey—Integrated Survey of Agriculture) and find that increasing dryness is associated with a higher probability of experiencing IPV [69].

However, in early 2020, two studies were published using multinational data from the Demographic and Health Surveys in sub-Saharan Africa (SSA) [13, 14], with contradictory findings. One paper by Cools et al. tested for an association between drought and physical IPV

using three different methodologies on a data set of 149,032 women from 17 SSA countries, and found no evidence of strong association between drought and IPV [14]. The second study by Epstein et al. used data on 83,990 women across 19 SSA countries. They found strong associations between drought and the likelihood of having a controlling partner as well as experiencing physical and sexual IPV [13].

Both studies estimated logistic regression models, but with different specifications. Cools et al. controlled for spatial autocorrelation to account for non-random exposure to drought since women who live in nearby areas are more likely to be exposed to similar drought conditions at a given point in time compared to women who live in distant areas. Epstein et al., on the other hand, controlled for household and individual factors that may affect a woman's risk of IPV, such as age, marital status, number of births, household size, and partner's education. Controlling for these local factors allowed them to estimate the contribution of drought to the variance in IPV with more precision.

Both studies also used various methods to ensure the robustness of their findings, such as including survey-level fixed effects and defining drought conditions based on long-term norms at each site, to ensure that the occurrence of droughts in their analyses would not be correlated with natural aridity. Both emphasized that the size of their samples, with tens of thousands of respondents from multiple countries, gave them more statistical power. Finally, both papers sought to improve the robustness of their findings with additional analyses. Epstein et al. ran country-specific analyses as well as pooled models that sequentially left out one country [13]. On the other hand, Cools et al. used multiple rounds of cross-sectional data per country for grid-level pseudo-panel analyses, and used age of first IPV experience for an event-history analysis [14]. Both authors found that, overall, their subsequent analyses supported their base-case findings.

Our objective in this paper is to resolve the contradictory findings put forth in both papers. Thus, we synthesize the methods used by each and draw on an even larger data set from SSA than those used in the two studies, as well as analyze data from two other developing regions—LAC and Asia. Finally, under the premise that the effects of drought on IPV may be context-specific, we investigate if there are strong effects observable at at the country level by running models for each country in our dataset that had sufficient observations during both drought and non-drought conditions.

## Materials and methods

### Data sources

Similar to previous researchers, we draw on data from Demographic and Health Surveys (DHS), which contain geolocated data from developing countries around the world provided by the US Agency for International Development, or USAID [70]. Since 2000, the DHS has included a domestic violence module that many countries have chosen to implement, which asks married women about their experience of various forms of intimate-partner violence. For the module, only one woman between the ages of 15 and 49 is selected at random from each household. Women who had male "partners," defined as men that they live or have lived with as if married [71], are asked detailed questions about their experience of violence perpetrated by that partner. The protocols for selecting and interviewing women are aimed at increasing the disclosure of actual violence while still ensuring women's safety [72]. We draw on 63 DHS surveys from SSA, LAC, and Asia, to assemble a data set of 363,428 women from 40 countries from the years 2000 until 2018 (Fig 1). Women with missing data for any of the variables used in the analysis were excluded from the study.

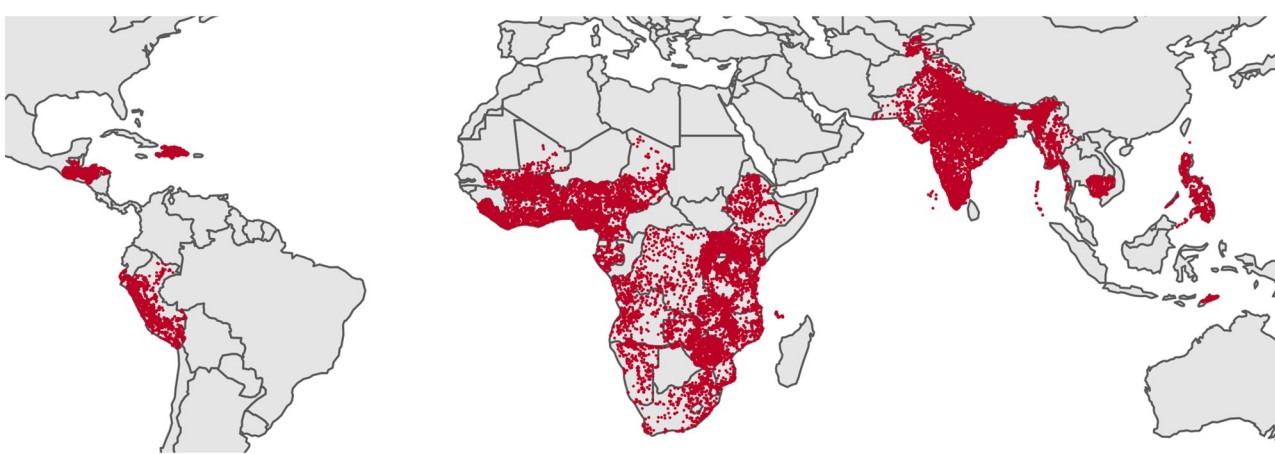

**Fig 1. Study locations.** Location of every DHS cluster included in the study.

To estimate drought severity, we use the Climate Hazards Group Infra-Red Precipitation with Stations (CHIRPS) [73] data set, which has gridded precipitation data from 1981 to the present and was used in the Epstein et al. study [13]. CHIRPS is a reanalysis data set generated by combining ground stations, satellite observations, and mathematical models to estimate rainfall across the world. The global reanalysis allows us to estimate levels of drought even in developing countries with sparse on-the-ground rainfall observations.

## Empirical strategy

To assess the association between drought and IPV, we aim to synthesize the methodologies used by Epstein et al. and Cools et al. [13, 14]. We run logistic regressions with a woman's report of experiencing various forms of IPV as the outcome variable, with prevailing drought conditions over the previous year as the independent variable of interest. A summary comparing the methods used by Epstein et al., Cools et al., and this paper is given in Table 1.

For each DHS cluster, we determine the baseline distribution of annual rainfall, and then estimate the percentile for the 12 months before the survey, the reference period for the IPV data. Drought conditions are classified as moderate when rainfall was 10%—30% of normal, severe when it was 2.5%—10%—2.5% of normal, and extreme when rainfall was 2.5%—0% of normal. This combines the two methods used by Epstein et al., who used cutoffs at 30% and 10%, and Cools et al., who used cutoffs at 10% and 2.5%. Examining the proportion of each

**Table 1. Comparison of methods used between Cools et al., Epstein et al., and this analysis.**

|  | Cools et al. | Epstein et al. | This Analysis |
|---|---|---|---|
| Scale | SSA (29 surveys) | SSA (19 surveys) | SSA (40 surveys) |
|  |  |  | Asia (12 surveys) |
|  |  |  | LAC (10 surveys) |
| Controlled for Spatial Autocorrelation? | **Yes** | No | **Yes** |
| Controlled Individual-Level Variables? | No | **Yes** | Yes |
| Examined Multiple types of IPV? | No | **Yes** | Yes |
| Cutoffs for Drought | 10 and 2.5% | 30 and 10% | 30, 10 and 2.5% |
| Survey Fixed Effects? | **Yes** | **Yes** | **Yes** |

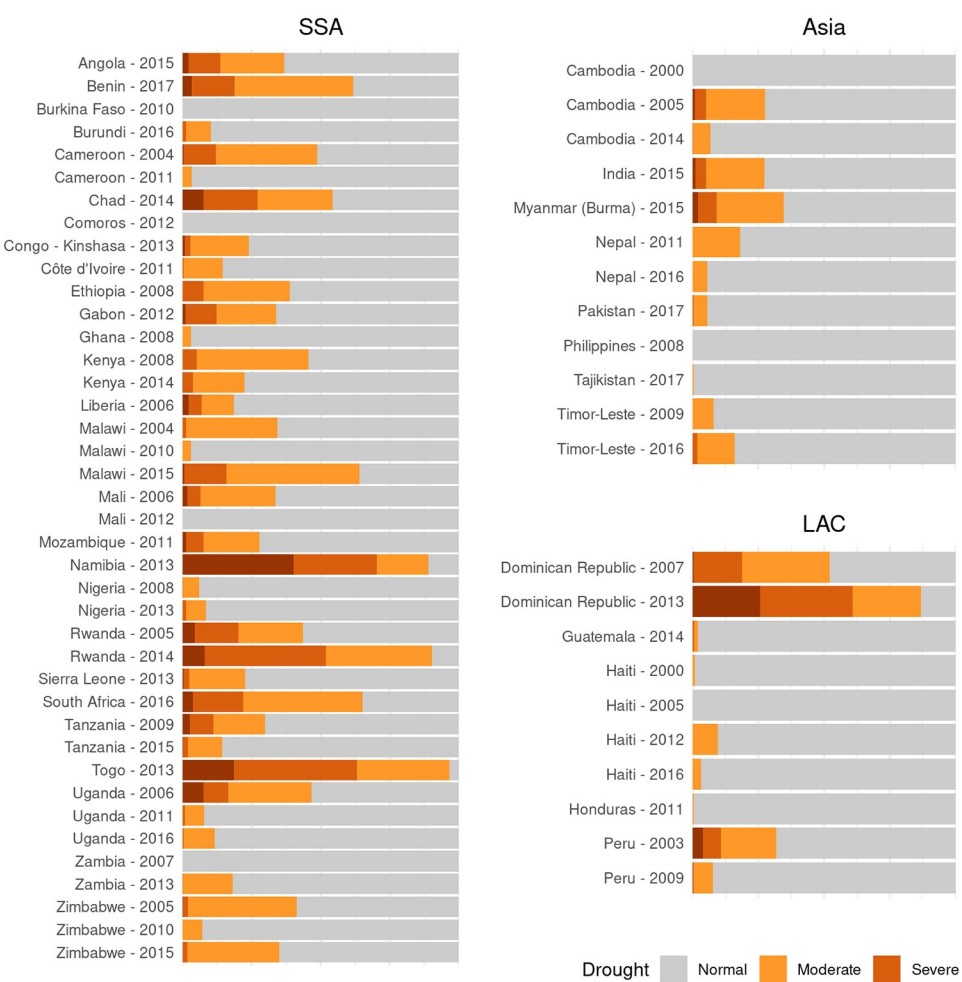

**Fig 2. Drought levels.** Proportion of each DHS survey exposed to each level of drought severity.

DHS survey in each of these categories shows that multiple surveys capture each level of drought severity across each world region (Fig 2).

Similar to Epstein et al., we include individual and household variables known or hypothesized to correlate with rates of IPV. We include the same variables as Epstein et al. and categorize the variables the same way, including whether the woman was married to her partner, whether the woman was literate, the woman's age, the number of births she'd had, her partner's education, her partner's age, the household size, as well as whether or not the household was classified as urban or rural by the DHS. In cases where women had missing or flagged data for these variables, they were removed from the analysis.

Additionally, following the approach of Epstein et al., we look at four types of intimate-partner violence: *physical violence*, which includes a respondent being pushed, shook, or having something thrown at her, being slapped, being punched or hit with something harmful, being kicked or dragged, being strangled or burnt, being threatened by a knife or gun, as well as if she ever had her arm twisted or hair pulled; *sexual violence*, which includes being forced into unwanted sex or sex acts; *emotional violence*, which includes being humiliated, threatened, insulted, or made to feel bad; and *controlling behaviors*, which include jealousy from the partner if the respondent talks to other men, accusations of unfaithfulness, not permitting the

**Table 2. Summary of variables used in regressions.**

| | | Africa | | Asia | | LAC | |
|---|---|---|---|---|---|---|---|
| | | *n* | % | *n* | % | *n* | % |
| IPV | Controlling | 129,023 | (66.2%) | 45,089 | (44.8%) | 44,568 | (65.6%) |
| | Emotional | 40,389 | (20.7%) | 11,688 | (11.6%) | 12,476 | (18.4%) |
| | Physical | 37,782 | (19.4%) | 19,107 | (19%) | 8,304 | (12.2%) |
| | Sexual | 17,415 | (8.9%) | 4,904 | (4.9%) | 3,350 | (4.9%) |
| Drought | No Drought | 147,425 | (75.7%) | 80,058 | (79.5%) | 53,754 | (79.1%) |
| | Moderate | 34,488 | (17.7%) | 16,873 | (16.8%) | 8,250 | (12.1%) |
| | Severe | 10,068 | (5.2%) | 2,947 | (2.9%) | 4,013 | (5.9%) |
| | Extreme | 2,839 | (1.5%) | 769 | (0.8%) | 1,944 | (2.9%) |
| Woman is Married | | 163,892 | (84.1%) | 98,927 | (98.3%) | 33,569 | (49.4%) |
| Woman is Literate | | 80,852 | (41.5%) | 58,530 | (58.2%) | 50,582 | (74.4%) |
| Woman's Age | < 20 | 13,697 | (7%) | 2,967 | (2.9%) | 3,722 | (5.5%) |
| | 20–29 | 81,994 | (42.1%) | 35,236 | (35%) | 23,202 | (34.1%) |
| | 30–39 | 64,737 | (33.2%) | 38,557 | (38.3%) | 25,078 | (36.9%) |
| | 40–49 | 34,392 | (17.7%) | 23,887 | (23.7%) | 15,959 | (23.5%) |
| Number of Births | 0–2 | 61,515 | (31.6%) | 45,792 | (45.5%) | 28,959 | (42.6%) |
| | 3–4 | 55,997 | (28.7%) | 32,099 | (31.9%) | 21,210 | (31.2%) |
| | > 4 | 64,321 | (33%) | 14,478 | (14.4%) | 13,424 | (19.8%) |
| Partner Education | None | 58,403 | (30%) | 16,956 | (16.8%) | 5,623 | (8.3%) |
| | Primary | 65,002 | (33.4%) | 19,979 | (19.9%) | 28,772 | (42.3%) |
| | Secondar | 57,374 | (29.4%) | 48,729 | (48.4%) | 25,549 | (37.6%) |
| | Higher | 14,041 | (7.2%) | 14,983 | (14.9%) | 8,017 | (11.8%) |
| Partner Age | < 20 | 804 | (0.4%) | 507 | (0.5%) | 610 | (0.9%) |
| | 20–29 | 40,254 | (20.7%) | 20,699 | (20.6%) | 15,922 | (23.4%) |
| | 30–39 | 71,466 | (36.7%) | 39,502 | (39.2%) | 25,257 | (37.2%) |
| | 40–49 | 49,534 | (25.4%) | 28,692 | (28.5%) | 17,930 | (26.4%) |
| | > 49 | 32,762 | (16.8%) | 11,247 | (11.2%) | 8,242 | (12.1%) |
| Household Size | 1–3 | 42,370 | (21.7%) | 17,867 | (17.8%) | 16,083 | (23.7%) |
| | 4–5 | 67,845 | (34.8%) | 43,991 | (43.7%) | 30,275 | (44.5%) |
| | > 5 | 84,605 | (43.4%) | 38,789 | (38.5%) | 21,603 | (31.8%) |
| Household is Rural | | 134,680 | (69.1%) | 68,781 | (68.3%) | 35,577 | (52.3%) |
| Total | | 194,820 | | 100,647 | | 67,961 | |

respondent to contact her family, and not permitting the respondent to meet with female friends. For sexual, physical, and emotional violence, women are asked about their experiences specifically over the previous year; for controlling behaviors, they were only asked about if their partner exhibited those behaviors. Finally, in some cases, additional questions were added between the 6th and 7th versions of the DHS, and these questions were included in determining whether a woman experienced each form of IPV.

All of these variables, for the three world regions we study, are summarized in Table 2.

By including individual and household variables in the manner of Epstein et al. and including a spatial term in the manner of Cools et al., we arrive at Eq 1, which was fit for each of the four types of IPV, and for each continent.

$$\log\left(\frac{p}{1-p}\right) = \beta_0 + \beta X + \beta_d x_d + \delta + s(lat, lon) + \epsilon \tag{1}$$

Where $p$ is the probability of a woman reporting an experience of a given form of IPV; $\beta_0$ is an

intercept; $\beta$ is a vector of coefficients modifying a matrix $X$ of covariates related to the woman, her partner, and her household; $\beta_d$ is the coefficient of interest, for prevailing drought conditions $x_d$; $\delta$ is a vector of fixed effects at the survey level; $s(lat, lon)$ is a spatial term that is a function of the latitude *lat* and longitude *lon* of each observation, and $\epsilon$ is a stochastic error term.

Following the methodology of Cools et al., we aim to control for spatial autocorrelation in our models. Spatial autocorrelation is a common source of of Type I errors and occurs when two observations that are nearby spatially also have similar values. Thus, they are not treated at random. Both drought and IPV occur with specific spatial patterns, and the probability that a drought and IPV are associated by coincidence is greater than the probability of a coincidental association if there were no spatial pattern to the data, making a hypothesis test that assumes independence invalid. For a larger discussion of spatial autocorrelation and Type I errors specifically in the context of using meteorological data see [74].

There are many ways to control for spatial autocorrelation in a regression. Common approaches involve calculating a spatial lag or a spatial error term based on a spatial weights matrix [75]. However, because the spatial weights matrix grows exponentially with the size of the data, it is not computationally feasible in an analysis like this one that draws on hundreds of thousands of observations. Rather, we use spatial splines, which can control for spatial autocorrelation and do not require a spatial weights matrix [76]. Specifically, we use thin-plate splines on the sphere [77] to estimate the spatial term $s(lat, lon)$ in our model. A key parameter in determining the complexity of thin-plate splines is the number of knots to include. We fit increasingly complex splines with 50, 100, 500, 1000, and 1500 knots. We ran our models with spatial splines using the `mgcv` package in R [78].

To determine if a model is properly specified and has controlled for spatial autocorrelation, one conducts a Moran's I test on the model residuals [79]. Similar to a spatial error or spatial leg regression, a Moran's I test requires a spatial weights matrix, which also causes problems for large datasets [80]. Thus, to conduct the test on hundreds of thousands of observations, we developed a software package in R and C++ for calculating the Moran's I test statistic that calculates the spatial weights matrix on-the-fly, dramatically reducing the amount of computer memory required [81]. For each continent and type of IPV, we select as our best model the one with the fewest number of spline knots that has no statistically significant autocorrelation in the residuals at $\alpha = 0.05$.

Finally, to investigate whether there is a relationship between drought and IPV for groups of women that are potentially more drought affected, we conduct analyses for all women in various categories related to household wealth, the woman's employment, household water sources, urbanization. We use the same model as in Eq 1, and similarly include increasingly complex thin-plate splines on the sphere until there was no spatial autocorrelation in the residuals. To simplify the analysis and reporting, we discretized the precipitation percentiles into two categories: drought ($< 30\%$ of normal) and normal ($> 30\%$ of normal). For urbanization, we use categories provided by the DHS; for women's employment and household water sources, we harmonize survey specific categories; for household wealth, we harmonize the survey-specific wealth factors using the methodology described by a DHS methodological report [82], then we take wealth quintiles across all surveys.

In our results, we report our coefficients as an average marginal effect, which is the average change in percentage points in a woman's probability of experiencing a given form of IPV. In our model, the effect of each level of drought is in reference to a no-drought baseline. Additionally, we report p-values using Bonferroni correction to reduce the risk of false positives. Thus, for our analysis of four outcomes across three world regions, for an $\alpha$ of 0.05, we use the value 0.05/12, since we are testing 12 hypothesis.

All code necessary to reproduce our results is available on github.

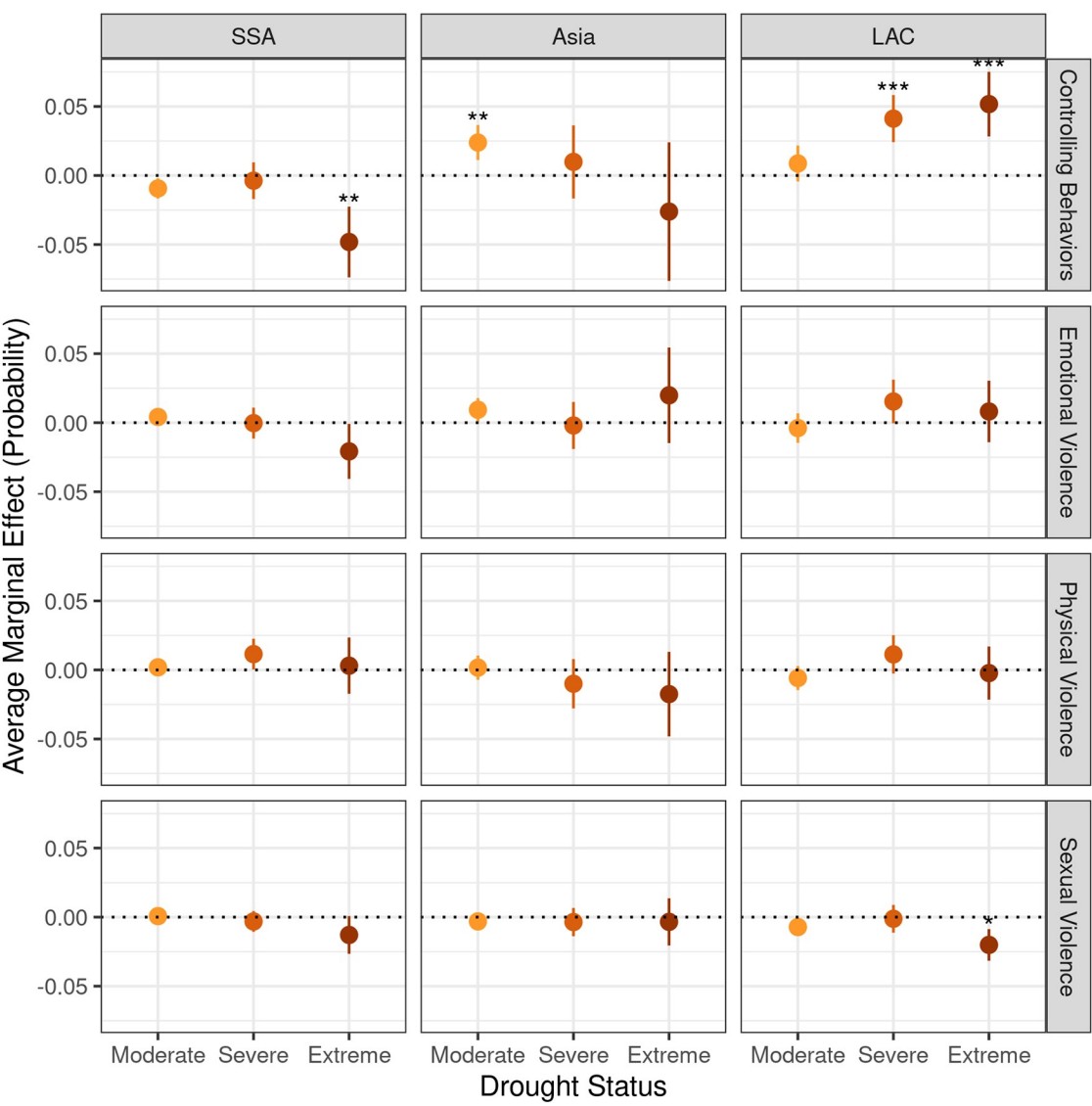

**Fig 3. Effects of droughts on IPV.** Average marginal effect of a moderate, severe, or extreme drought on a woman's probability of experiencing IPV, relative to non-drought conditions, for all four types of IPV across three different sized data sets. Significance values are Bonferroni-corrected for testing 12 hypotheses, so $^*p < 0.05/12$, $^{**}p < 0.01/12$, $^{***}p < 0.001/12$.

## Results

Our models showed no significant relationship between any level of drought and emotional or physical violence on any continent (Fig 3). Additionally, in Latin America, extreme drought was associated with slightly less sexual violence.

While emotional, physical, and sexual violence had no clear association with drought, drought had a strong association (in both magnitude and statistical significance) with controlling behaviors on all three continents. In Asia and LAC, drought was associated with greater rates of women reporting controlling behaviors. In LAC, for example, a woman's probability of reporting a partner with controlling behaviors increases about 4.1 percentage points during severe drought and 5.1 percentage points during an extreme drought. However, the converse

**Table 3. Effect of drought on IPV.**

| Variable | Category | Controlling | Emotional | Physical | Sexual |
|---|---|---|---|---|---|
| Wealth | Middle | 0.012 | 0.01 | 0.008 | 0.004 |
| | Poorer | -0.005 | -0.001 | 0.001 | -0.002 |
| | Poorest | 0.005 | -0.003 | -0.008 | -0.005 |
| | Richer | -0.005 | 0.012 | -0.001 | -0.007 |
| | Richest | -0.001 | -0.004 | -0.003 | -0.001 |
| Employment | Agriculture | 0.001 | -0.001 | 0 | -0.002 |
| | Manual Labor | -0.008 | -0.004 | -0.008 | -0.006 |
| | Professional | 0.008 | 0.008 | 0.002 | -0.002 |
| | Unemployed | 0.011 | -0.079 | 0.017 | -0.011 |
| Water Source | Dug Well Water | -0.002 | 0.014 | 0.01 | 0.003 |
| | Other | -0.008 | -0.062 | -0.07 | 0.009 |
| | Piped Water | 0.011 | 0.004 | 0.001 | -0.007* |
| | Purchased Water | -0.01 | -0.003 | 0.001 | -0.007 |
| | Rain Water | -0.038 | -0.036 | -0.021 | 0.001 |
| | Surface Water | 0.021 | 0.003 | 0.002 | 0.003 |
| | Tube Well Water | -0.006 | 0 | -0.008 | -0.007 |
| Urbanization | Rural | 0.001 | 0.004 | -0.001 | 0 |
| | Urban | 0.001 | 0.009 | 0.005 | -0.005 |

Average Marginal Effect (AME) of drought on a woman's probability of experiencing IPV, for women in different categories of wealth, employment, drinking water source, and urbanization. Significance values are Bonferroni-corrected for testing 4 hypotheses, so $^*p < 0.05/4$, $^{**}p < 0.01/4$, $^{***}p < 0.001/4$).

was true in sub-Saharan Africa: extreme drought was associated with women being less likely to report controlling behaviors.

For the sake of comparability, we summarize results for models with no spatial terms in the supplement. In models that did not control for spatial autocorrelation, statistically significant effects were much more common, even after Bonferonni correction, although drought was associated with more IPV about as often as it was associated with less IPV. Comparing models fit with and without spatial terms shows that, in all cases, including spatial terms improved the AIC of the models. While including spatial terms changed the model estimates for parameters related to drought, the estimates for parameters at the individual and household level were quite similar in terms of magnitude and significance. Finally, we ran analysis for different sub-groups of women to test for associations in contexts where women are potentially more drought affected. This analysis found found that women relying on piped water were 0.7 percentage points less likely to report experiencing sexual violence in the previous year. There were no significant associations between drought and women reporting experiencing IPV at different levels of urbanization, in different employment categories, at different wealth quintiles, or for household water sources other than piped water (Table 3).

## Discussion

Two recent papers by Cools et al. [14] and by Epstein et al. [13] found apparently contradictory results concerning the relationship between drought and IPV. We attempted to resolve this contradiction by replicating their analyses with a synthesis of their methods and using a much larger data set with surveys from multiple continents. We found no observable relationship between drought and most forms of IPV at continental scales, although we found evidence that droughts do affect controlling behaviors in all of the continents we analyzed.

We took great care to control for spatial autocorrelation in this study. By our evaluation metrics of AIC and the Moran's I test of spatial autocorrelation, the models with spatial terms were a much better fit than the models with no spatial terms [79]. This suggests that there is a major spatial component in the distribution of drought and IPV, and that many of the findings of associations by Epstein et al., which did not control for spatial-autocorrelation, were Type I errors that incorrectly rejected a true null hypothesis [74]. This is further supported by the fact that including more datasets but running models with the same form that Epstein et al. used (S1 Fig in S1 File) showed many significant relationships between drought and IPV in sub-Saharan Africa, but found more cases of drought having a protective effect on IPV than drought increasing the risk of IPV.

Our best models showed little association between drought and emotional, physical, and sexual violence on any continent, yet we found associations between drought and increases in controlling behaviors in LAC and Asia. The effect of drought on controlling behaviors is supported by literature that finds that controlling behaviors are associated with precarious economic situations and unemployment [83]. However, given that controlling behaviors are a risk factor for other forms of violence in other countries [84, 85], it is surprising that drought can increase the risk of controlling behaviors without also affecting the risk of other forms of violence. Some authors have emphasized that different forms of IPV have different etiologies [86, 87], and our results suggest that this is also true in a global context.

While drought is associated with controlling behaviors on all three continents, extreme drought actually has a protective effect against controlling behaviors in sub-Saharan Africa. This may be due to coping strategies specific to that continent that separate the genders [5]. As the most arid and least irrigated continent in our study, drought in Africa may induce individuals to spend more time away from their partner in order to travel further distances to obtain water or leave the household to pursue off-farm labor opportunities to a greater degree than in Asia or LAC, and this may explain why controlling behaviors actually decrease with extreme drought.

It should also be noted that controlling behaviors are the only form of IPV in our dataset that were not specifically asked about over the previous 12 months, the period for which we measured drought. Rather, women were simply asked whether their partner exhibited those behaviors or not. Thus, this analysis rests on the assumption that women are more likely to report a controlling partner if their partner has exhibited those behaviors in the previous twelve months. While this is a strong assumption, we did find statistically significant effects of drought on controlling behaviors in all three continents, and we fully controlled for false positives due to spatial autocorrelation. This implies that women's reporting of controlling behaviors is influenced by her recent experiences with her partner, and this is a suitable variable to examine in relation to drought.

It could be argued that reducing the risk of a Type I error by controlling for spatial autocorrelation puts our analysis at risk of a Type II error, of failing to reject the null hypothesis and claiming there is no effect of drought on IPV when in fact one does exist. While the estimated coefficients for the drought terms were greatly diminished between the models with no spatial terms and the models with spatial terms (Fig 3), many other variables related to IPV, such as the woman's level of education, remained significant even as spatial autocorrelation was controlled for. This suggests that the models are still capturing real effects, and the drought terms were the most affected by the problem of spatial autocorrelation [79].

The impacts of drought are not borne evenly across households. Rather, households that are poor, rural, engaged in agricultural livelihoods, or dependent on rainfed water sources are much more likely to be affected by drought. Households that are already water insecure in particular are much more likely to be affected by drought [27, 28]. Thus, we subset our analysis to

women in different classes of potentially higher vulnerability to test whether a significant association could be observed between drought and different forms of IPV. We found little association between drought and any for of IPV for any class of household. Even our findings around controlling behaviors at the continental scale were not present for subsets of the population, even subsets that are likely to be drought-affected. This calls into question the robustness of our findings around drought and controlling behaviors.

As discussed in the introduction, there are strong theoretical reasons to hypothesize a linkage between drought and IPV, including emotional, sexual, and physical violence [32, 33]. There are many examples of drought having a more adverse impact on women than men [9], and women with less economic standing than their partners are in many cases more vulnerable to IPV [88]. Moreover, drought has been associated with a variety of other forms of violence, from civil unrest to crime [51, 53]. The fact that there are strong theoretical reasons to suspect a linkage between drought and IPV yet no empirical evidence of a link in our analysis outside of controlling behaviors suggests that the effect may be more localized.

The purpose of this analysis was to examine the contradictory conclusions of Epstein et al. and Cools et al. by integrating their methods and using an expanded dataset. Thus, we used the definitions of drought shared by both researchers based on rainfall percentiles in the year before the DHS survey, as well as the covariates used by Epstein et al. and controlling for spatial autocorrelation in the manner of Cools et al. Our analysis showed that, using this definition of drought and these methodological approaches, there is no evidence of an association between drought and forms of IPV outside of controlling violence. However, drought can operate on many different timescales and can influenced by temperature well as precipitation. Thus, this work should not be interpreted as evidence that there is no relationship between drought and IPV, but just that there is no relationship under the definition of drought currently used in the literature. Future work should explore different ways to measure drought, accounting for different timescales and hydrological processes.

## Conclusion

Our analysis shows the importance of controlling for spatial autocorrelation in studies using data that is not randomly distributed in space, such as climatological and meteorological data, especially for the purposes of hypothesis testing. In this paper we demonstrated a method of using spatial splines to control for spatial autocorrelation and a Moran's I test to evaluate the autocorrelation in the residuals. All of our models with spatial splines performed better than our models without spatial splines, indicating that many of the effects observed in the models without spatial splines were false positives. Taking this approach and controlling for spatial autocorrelation also shows that our findings that drought can affect the prevalence of controlling behaviors are robust.

## Supporting information

**S1 File.**
(PDF)

## Author Contributions

**Conceptualization:** Matthew Cooper, Sveva Vitellozzi, Yeyoung Lee, Greg Seymour, Carlo Azzari.

**Data curation:** Matthew Cooper.

**Formal analysis:** Matthew Cooper.

**Investigation:** Matthew Cooper, Austin Sandler, Sveva Vitellozzi, Yeyoung Lee, Greg Seymour, Beliyou Haile.

**Methodology:** Matthew Cooper, Austin Sandler, Carlo Azzari.

**Project administration:** Matthew Cooper, Carlo Azzari.

**Resources:** Matthew Cooper.

**Software:** Matthew Cooper.

**Supervision:** Matthew Cooper.

**Validation:** Matthew Cooper, Yeyoung Lee.

**Visualization:** Matthew Cooper.

**Writing – original draft:** Matthew Cooper, Sveva Vitellozzi, Greg Seymour.

**Writing – review & editing:** Matthew Cooper, Austin Sandler, Beliyou Haile, Carlo Azzari.

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
