## [Decision Letter · Decision Letter 0]

28 May 2021

PONE-D-21-11426

Reexamining the Effects of Drought on Intimate-Partner Violence

PLOS ONE

Dear Dr. Matthew,

Thank you for submitting your manuscript to PLOS ONE. After careful consideration, we feel that it has merit but does not fully meet PLOS ONE’s publication criteria as it currently stands. Therefore, we invite you to submit a revised version of the manuscript that addresses the points raised during the review process.

We look forward to receiving your revised manuscript.

Kind regards,

Shah Md Atiqul Haq

Academic Editor

PLOS ONE

Additional Editor Comments:

Dear Authors,

Thank you for submitting this very interesting and important paper.

Based on the comments received, I suggest a minor revision.

Please revise the paper and resubmit it for further editing.

Good luck

Journal Requirements:

2. In the Methods section or as supplementary material please clearly reference the source  of 63 DHS data used as a part of the study.

"NO"

5. We note that Figure 1 in your submission contain map images which may be copyrighted. All PLOS content is published under the Creative Commons Attribution License (CC BY 4.0), which means that the manuscript, images, and Supporting Information files will be freely available online, and any third party is permitted to access, download, copy, distribute, and use these materials in any way, even commercially, with proper attribution. For these reasons, we cannot publish previously copyrighted maps or satellite images created using proprietary data, such as Google software (Google Maps, Street View, and Earth). For more information, see our copyright guidelines: http://journals.plos.org/plosone/s/licenses-and-copyright.

5.1.    You may seek permission from the original copyright holder of Figure 1 to publish the content specifically under the CC BY 4.0 license. 

5.2.    If you are unable to obtain permission from the original copyright holder to publish these figures under the CC BY 4.0 license or if the copyright holder’s requirements are incompatible with the CC BY 4.0 license, please either i) remove the figure or ii) supply a replacement figure that complies with the CC BY 4.0 license. Please check copyright information on all replacement figures and update the figure caption with source information. If applicable, please specify in the figure caption text when a figure is similar but not identical to the original image and is therefore for illustrative purposes only.

Reviewers' comments:

Reviewer's Responses to Questions

**Comments to the Author**

1. Is the manuscript technically sound, and do the data support the conclusions?

Reviewer #1: Yes

Reviewer #2: No

Reviewer #3: Yes

2. Has the statistical analysis been performed appropriately and rigorously? 

Reviewer #1: I Don't Know

Reviewer #2: N/A

Reviewer #3: I Don't Know

3. Have the authors made all data underlying the findings in their manuscript fully available?

Reviewer #1: Yes

Reviewer #2: No

Reviewer #3: Yes

4. Is the manuscript presented in an intelligible fashion and written in standard English?

Reviewer #1: Yes

Reviewer #2: No

Reviewer #3: No

5. Review Comments to the Author

Reviewer #1: The authors have taken a great subject to study and probably have put their best efforts in that. However, my observation is that somehow paper has lost the track of highlighting the relationship between drought and IPV and has focused much on establishing the statistical model/analyses to study the relationship between drought and IPV.

This paper has great potential to get publish however need to focus more on theoretical part of relationship of these two variables using the results of the study.

Reviewer #2: The authors attempt to try relationship between the effects of droughts on the risk of Intimate-Partner Violence (IPV).

My comments are as follows.

Comments:

• I have a fundamental problem with the article. This is an overly sophisticated article. Of course, we can look for a relationship between any variables, and we can also find significant correlations between them, but I can hardly imagine that there is a real physical relationship between IPV and the environmental variables associated with it.

Reviewer

Reviewer #3: The study is very interesting related to Effects of Drought on Intimate-Partner Violence.

I would like to suggest some points which would help in improving the manuscript.

abstract:

In line one instead of writing a number of , it may be written as severa.

Autocorrection word is togather and not separated as in abstract.

Introduction:

Lines 9-13 i.e from shocks related...... experiencing IPV is not very clear. Needs to be rewritten.

Introduction is loo long and need to be written short and crisp.

Discussion needs to be made more elaborate with references from other studies. Very little refrences seen in the part of discussion

6. PLOS authors have the option to publish the peer review history of their article (what does this mean?). If published, this will include your full peer review and any attached files.

Reviewer #1: No

Reviewer #2: No

Reviewer #3: **Yes: **Dr Anjali Somal

---

## [Author Response · Author response to Decision Letter 0]

24 Jun 2021

Dear Reviewers,

Thank you for your feedback and comments on our manuscript. We have made changes according to your suggestions, including focusing more on the theory behind the drought-IPV relationship, especially in the discussion section. We have also edited the wording and phrasing in the abstract and introduction section where requested.

Thank you again for your insightful and helpful feedback.

---

## [Editor Report · Decision Letter 1]

25 Jun 2021

Reexamining the Effects of Drought on Intimate-Partner Violence

PONE-D-21-11426R1

Dear Dr. Matthew,

We’re pleased to inform you that your manuscript has been judged scientifically suitable for publication and will be formally accepted for publication once it meets all outstanding technical requirements.

Kind regards,

Shah Md Atiqul Haq

Academic Editor

PLOS ONE

Additional Editor Comments (optional):

Dear Authors,

Thank you for responding to the reviewers' comments.

Congratulations!!!

The article is now accepted.

Best wishes for future research,
---

## [Editor Report · Acceptance letter]

12 Jul 2021

PONE-D-21-11426R1 

Re-examining the effects of drought on intimate-partner violence 

Dear Dr. Cooper:

I'm pleased to inform you that your manuscript has been deemed suitable for publication in PLOS ONE. Congratulations! Your manuscript is now with our production department. 

Kind regards, 

on behalf of

Dr. Shah Md Atiqul Haq 

Academic Editor

PLOS ONE